

# Current temporal asymmetry and the role of tides: Nan-Wan Bay vs. the Gulf of Elat

Yosef Ashkenazy[1], Erick Fredj[2], Hezi Gildor[3], Gwo-Ching Gong[4], and Hung-Jen Lee[5]

[1]Department of Solar Energy and Environmental Physics, BIDR, Ben-Gurion University, Midreshet Ben-Gurion, 84990, Israel
[2]The Jerusalem College of Technology, Jerusalem, Israel
[3]Institute of Earth Sciences, The Hebrew University, Edmond J. Safra Campus, Givat Ram, Jerusalem, 91904, Israel
[4]Institute of Marine Environment and Ecology, National Taiwan Ocean University, Keelung, Taiwan
[5]Department of Marine Environmental Informatics, National Taiwan Ocean University, Keelung, Taiwan

*Correspondence to:* Y. Ashkenazy (ashkena@bgu.ac.il)

**Abstract.** Nan-Wan Bay in Taiwan and the Gulf of Elat in Israel are two different coastal environments, and as such, their currents are expected to have different statistical properties. While Nan-Wan Bay is shallow, has three open boundaries, and is directly connected to the open ocean, the Gulf of Elat is deep, semi-enclosed, and connected to the Red Sea via the Straits of Tiran. Surface currents have been continuously measured with fine temporal (less than or equal to one hour) and spatial resolution (less than or equal to one km) for more than a year in both environments using coastal radars (CODARs) that cover a domain of roughly $10 \times 10$ kms. These measurements show that the currents in Nan-Wan Bay are much stronger than those in the Gulf of Elat and that the mean current field in Nan-Wan Bay exhibits cyclonic circulation, which is stronger in the summer; in the Gulf of Elat, the mean current field is directed southward and is also stronger during the summer. We have compared the statistical properties of the current speeds in both environments and found that both exhibit large spatial and seasonal variations in the shape parameter of the Weibull distribution. However, we have found fundamental and significant differences when comparing the temporal asymmetry of the current speed (i.e., the ratio between the time during which the current speed increases and the total time). While the Nan-Wan Bay currents are significantly asymmetric, those of the Gulf of Elat are not. We then extracted the tidal component of the Nan-Wan Bay currents and found that it is strongly asymmetric, while the asymmetry of tidally filtered currents is much weaker. We thus conclude that the temporal asymmetry of the Nan-Wan Bay currents reported here is due to the strong tides in the region. We show that the asymmetry ratio in Nan-Wan Bay varies spatially and seasonally: (i) the currents increase rapidly and decay slowly in the northern part of the domain and vice versa in the southern part, and (ii) the asymmetry is stronger during summer.

## 1 Introduction

Ocean variability covers a wide range of temporal and spatial scales, from seconds to tens of thousands of years and from millimeters to tens of thousands of kilometers. Obviously, even the most advanced ocean models cannot resolve this wide range of scales, and thus they use sub-grid parameterizations to account for such phenomena. Modeling of oceanic dynamics is



often based on forcing from point measurements and on long-term-mean measurements (e.g., monthly averages). In addition, models are often calibrated (tuned) and validated against these long-term-mean point measurements. Such calibration and validation is performed under the assumption that these measurements represent the spatial grid resolution of the ocean model.

One key variable of ocean dynamics is ocean currents. There are various ways to measure ocean currents, particularly ocean

surface currents. Point measurement tools include rotor-based devices and the acoustic Doppler current profiler (ADCP). Various types of drifters and floats can be used to approximate the currents. Satellite data that measure the sea surface height can be used to estimate geostrophic currents; these provide, on a daily basis, global scale surface current maps with resolutions of a few kilometers. Coastal radar (CODAR, see details below) systems are increasingly used to measure surface currents at finer spatial (from a few hundred meters) and temporal (half an hour and more) scales. Such rich and detailed data can be used

to analyze the statistical properties and surface currents.

The Weibull distribution was previously used to characterize the probability density function (PDF) of altimeter-based surface currents (e.g., Chu, 2008, 2009), and global maps of the shape and scale parameters of the Weibull distribution were constructed. A detailed statistical analysis of CODAR-based surface currents from the northern tip of the Gulf of Elat (Aqaba) was performed by some of us (Ashkenazy and Gildor, 2011); it was found that the shape and scale parameters of the Weibull

distribution significantly vary in this small area of 6 km × 10 km (Fig. 1b,d). Using a variant of a simple Ekman layer model, this spatial variability was attributed to the temporal variability of the local winds.

The first goal of the present study was to verify whether the spatial variability of the statistical properties of the surface currents reported in Ashkenazy and Gildor (2011) are unique to Elat or whether they exist in a quite different environment such as the Nan-Wan Bay of Taiwan (Fig. 1a,c). There are several differences between the Gulf of Elat and Nan-Wan Bay: (i)

Nan-Wan Bay is open to the ocean from three sides, while the Gulf of Elat is a semi-enclosed basin (with one open boundary); (ii) Nan-Wan Bay is directly connected to the world ocean (the South China Sea), while the Gulf of Elat is connected to the world ocean (the Indian Ocean) via two straits, to and from the Red Sea; (iii) the water depth in Nan-Wan Bay (in the study area, Fig. 1c) is relatively shallow compared with the depth of the Gulf of Elat (in the study area, Fig. 1d); and (iv) the currents of Nan-Wan Bay have a strong tidal component, while those of the Gulf of Elat do not. We find that the level of variability in

these two different basins is similar.

The second goal of the present study was to identify and quantify the surface currents' temporal asymmetry (i.e., the ratio between the time during which the current speed increases and the total time) in both Nan-Wan Bay and the Gulf of Elat. We found that the surface currents in Nan-Wan Bay are significantly asymmetric, while those of the Gulf of Elat are symmetric. We show here that the asymmetry of Nan-Wan Bay's currents is rooted in the strong tides of the bay, while the absence of

asymmetry in the Gulf of Elat is associated with the relatively weak tidal signal in this gulf.

The paper is organized as follows. We first describe the research area of Nan-Wan Bay (Sec. 2.1) and the Gulf of Elat (Sec. 2.2). We next describe the CODAR data, the statistical methods used to evaluate the parameters of the currents PDFs, the detiding procedure, and the measure for current temporal asymmetry (Sec. 3). We then compare the statistical properties of Nan-Wan Bay with those of the Gulf of Elat (Sec. 4). The results regarding the temporal asymmetry of the currents and their

relation to the tides are discussed in Sec. 4. A summary then follows (Sec. 5).



## 2   Study regions

### 2.1   Nan-Wan Bay

Nan-Wan ("wan" in Chinese means bay) Bay is located at the southernmost part of Taiwan (Fig. 1a,c). It is bounded by western (Mou-Bi-Tou) and eastern (O-Luan-Bi) capes in which the distance between them is ∼14 km. The bay is surrounded by the Taiwan Strait (from the west), the South China Sea (from the southwestern direction), the Luzon Strait (from the south), and the Pacific Ocean (from the east). The bay includes some seamounts that partially block it. The eastern side of the bay includes a shallow continental shelf (≈ 5 km wide); the western side of the bay has a very small continental shelf. The curved (parallel to the coast) channel is partially bounded by the southern seamount.

Nan-Wan is subject to monsoonal winds; these blow from the southwestern direction during summer and from the northeastern direction during late fall, winter, and early summer. Thus, during winter, the strong monsoon winds blow downhill to the south from the mountains of Nan-Wan towards the bay. During the fall, wind-driven currents are weak compared to the tidally driven currents; yet, wind-driven currents are strong during a few typhoon events. The north-flowing Kuroshio Current is adjacent to Taiwan from the east, but still, its influence on Nan-Wan Bay is not significant compared to the tidally driven currents. The tidal current speed at spring tides exceeds 2 m/s (Lee et al., 1997, 1999a, b).

The semidiurnal and diurnal tidal components of the currents in Nan-Wan are modulated by the spring-neap tidal cycle. These two are approximately equal in their magnitude (Lee et al., 1997). The temperature hardly drops around the neap tides, while during the spring tides, the temperature can suddenly drop by several degrees for several hours due to strong tidally magnified upwelling. Such a temperature drop can reach 10°C. For example, on November 24, 1988, the sea surface temperature dropped suddenly (within a few hours) by 10°C (from 24°C to 14°C), leading to a massive fish kill. A similar event occurred during July 2008.

### 2.2   The Gulf of Elat

The study region is located in the northern terminus of the Gulf of Elat/Aqaba, in the northeastern region of the Red Sea; it is a nearly rectangular, deep (∼700 m; Fig. 1d), and semi-enclosed basin. A desert mountain range surrounds the Gulf of Elat and steers the persistent northerly wind along its main axis (Berman et al., 2003; Afargan and Gildor, 2015). Several components affect the water circulation/currents in the gulf: winds, tides, and the thermohaline circulation. The semidiurnal peak, forced by the flux of water through the Straits of Tiran, dominates the weak tidal component (Genin and Paldor, 1998; Monismith and Genin, 2004; Manasrah et al., 2006; Carlson et al., 2012). The surface current in the study region often exhibits a complex (although spatially coherent) pattern, including eddies that cover a large part of the domain (Gildor et al., 2010; Ashkenazy and Gildor, 2009, 2011).

The gulf is almost blocked from the cold and dense water of the world ocean due to the shallow sill (137 m) between the Indian Ocean and the Red Sea (Bab el Mandeb) and the shallow sill (240 m) between the Red Sea and the Gulf of Elat (the Tiran Strait) (Genin, 2008). Thus, the water column in the gulf exhibits weak stratification and winter deep water formation caused by surface cooling and evaporation (Wolf-Vecht et al., 1992; Genin et al., 1995; Biton et al., 2008; Biton and Gildor,



2011a; Carlson et al., 2014). During February and March, temperature and salinity are almost uniform down to a depth of a few hundred metres (and sometimes down to the bottom); new stratification begins to form in March (Wolf-Vecht et al., 1992). The gulf is stratified in summer when an upper (∼200 m) warm layer overlies a homogeneous deeper layer (Biton and Gildor, 2011b).

The wind in the Elat region is northerly (with a small easterly component) during most of the year (Ashkenazy and Gildor, 2011; Afargan and Gildor, 2015). Strong southerly winds occur rarely during the winter, usually during southern storms. There is a strong diurnal cycle associated with the diurnal breeze cycle in the summer (Saaroni et al., 2004). On average, the wind is stronger during summer.

# 3  Methods

## 3.1  HF-radar-based currents

High frequency (HF) radar systems for surface current measurements (Barrick et al., 1985; Gurgel et al., 1999b), like the SeaSonde (Hodgins, 1994) and Wellen Radar (WERA; Gurgel et al., 1999a), have become popular in recent years and have mainly been used to study coastal circulation. These systems usually operate at a frequency of ∼24 MHz or lower, covering distances from several tens of kilometers up to more than a hundred kilometers, at a resolution of a few kilometers.

Many articles have described in detail the theory behind the HF radar surface current measurements (e.g., Gurgel et al., 1999b; Barrick et al., 1985). Briefly, surface gravity waves reflect radio waves that were transmitted by the HF radar, and these are again detected by the radar. Surface currents can be measured based on the Bragg resonance of the surface waves with the transmitted radio waves. The received spectrum is not identical to the transmitted spectrum due to the Doppler shift caused by the radial component of the phase shift of incoming and outgoing waves. Waves that are superimposed on a current lead to a further shift in the spectral peaks. This additional shift allows the extraction of the radial component of the current. It is possible to calculate the surface velocity field based on two radar sites that measure the radial velocity of a patch of water from two different angles.

CODAR systems have been used to measure surface current fields both in Nan-Wan (Fig. 2a) and in the Gulf of Elat (see, e.g., Gildor et al., 2009, 2010). There are two CODAR stations in Nan-Wan (indicated by the red dots in Fig. 2a) operated at 24-27 MHz; a third CODAR station has been operated since June 2014 and is not included here. The CODARs produce current fields with a spatial resolution of 1.5 km and a temporal resolution of 1 hour. The CODAR measurements in the Gulf of Elat are conducted by two 42-MHz SeaSonde HF radar systems that were installed in the gulf in August 2005. They measure currents at a temporal resolution of half an hour and a spatial resolution of about 300 m. Below we analyze one year of surface current fields for both Nan-Wan (from March 1, 2013) and the Gulf of Elat (from October 1, 2005). The surface current fields were filtered and interpolated to fill spatial gaps in the observation (see, e.g., Lekien and Gildor, 2009; Lekien et al., 2004). There were many missing days of Nan-Wan Bay data from September to November, 2013, and we thus do not present the results of these autumn months. Consequently, the presented annual mean results underestimate the effect of the fall season.



## 3.2 The Weibull distribution

In a previous study (Ashkenazy and Gildor, 2011), some of us have analyzed the parameters of the Weibull distribution de-
scribing the PDF of the surface currents in the Gulf of Elat. It was found that these parameters exhibit a large spatial variability
that changes seasonally. Here we applied the same procedure on the Nan-Wan CODAR surface current measurements. We thus
only briefly describe the parameter estimation procedure, and the interested reader can find more details in Ashkenazy and
Gildor (2011).

The Weibull distribution was suggested as an appropriate PDF of wind (e.g., Monahan, 2006, 2010) and surface current
(e.g., Chu, 2008, 2009) speed. While it is possible that other distributions may be more appropriate models for current speeds,
we restricted ourselves to the Weibull distribution to allow comparison with previous results.

The Weibull PDF has two parameters, the scale and shape parameters, $\lambda$ and $k$, and is given by

$$f(x; \lambda, k) = \frac{k}{\lambda} \left(\frac{x}{\lambda}\right)^{k-1} e^{-(x/\lambda)^k}, \tag{1}$$

where $\lambda$ and $k$ are greater than zero. Given a dataset (in our case, the time series of surface current speed), it is possible to
estimate $k$ using (i) the different moments, (ii) the hazard function, and (iii) the maximum likelihood estimator of the Weibull
distribution. Once the shape parameter, $k$, is found, it is possible to estimate the scale parameter, $\lambda$, for example, by using the
relation between $\lambda$ and the mean of the time series. The different methods yielded similar results, and we thus present below
only the results that are based on the different moment estimation (see Ashkenazy and Gildor, 2011).

A typical current speed time series from Nan-Wan Bay (from the location marked by "×" in Fig. 2) is shown in Fig. 3a.
The corresponding PDF and the Weibull fit are shown in Fig. 3c,d. The estimated shape parameter is $k \approx 2$, indicating that
this specific PDF is close to the Rayleigh distribution. For comparison, we also plotted the Weibull distribution with $k = 1$,
which corresponds to the exponential distribution. Unlike the PDF of the data, this PDF decreases monotonically and indicates
a higher probability for high current speed values.

## 3.3 Detiding

To study the effect of tides on the CODAR currents, we implemented the algorithm of Pawlowicz et al. (2002). We decomposed
the current speed time series into the tidal component and detided time series. An example of such time series from Nan-Wan
Bay is shown in Fig. 3a,b. It is clear that the tidal component dominates the current time series, as reported by previous studies
on Nan-Wan Bay (Lee et al., 1997, 1999a, b).

## 3.4 Temporal asymmetry

To measure the temporal asymmetry of the current speed time series, we computed the ratio between the increasing speed
time steps and the total number time steps. This and similar measures were used to quantify the asymmetry of the temperature
time series (Bartos and Janosi, 2005; Ashkenazy et al., 2008). The asymmetry measure of the current speed time series $s_i$



($i = 1 \ldots N$ where $N$ is the length of the time series) can be expressed as:

$$A(\tau) = \frac{1}{N - \tau} \sum_{i=1}^{N-\tau} \Theta(s_{i+\tau} - s_i), \tag{2}$$

where $\Theta(x) = 1$ for $x > 0$ and is zero otherwise and $\tau$ is a time interval. Thus, when the number of positive increments is equal to the number of negative increments, $A = 0.5$. $A > 0.5$ ($A < 0.5$) indicates that the current speed increases (decreases) gradually and decreases (increases) rapidly. $2A - 1$ indicates by how much the number of positive increments exceeds the negative ones; for example, if $A = 0.55$, there are 10% more positive increments than negative increments. It is possible to measure the asymmetry over a different time interval $\tau$. When the time series is asymmetric and periodic, the asymmetry measure (2) will change sign when $\tau$ exceeds half of the period of the time series.

We used a surrogate data test to assess the significance of the asymmetry results. Specifically, we randomly shuffled the time series and then measured the asymmetry $A(\tau)$. If the original asymmetry lies outside the error bar of the surrogate data (e.g., outside the range of the mean $\pm$ 1 std.), the asymmetry of the original time series may be considered as significant.

## 4 Results

We first calculated the mean surface current field for each grid point in study area of Nan-Wan Bay (Fig. 2). Consistent with previous studies in the region (Lee et al., 1997, 1999a, b), the surface currents are strong and exhibit cyclonic circulation with the center located close to middle of the line connecting the capes of Nan-Wan Bay (Fig. 2b). The currents are much stronger during the summer (Fig. 2d) than during the winter (Fig. 2c). The mean current speed (in the research area) is 80 cm s$^{-1}$ in the summer and 60 cm s$^{-1}$ in the winter (Table 1). Moreover, as is reflected in Fig. 2, the mean zonal velocity during the summer is more than four times greater than the winter one (Table 1), pointing to an eastward current south of the cyclonic eddy. The current speed is relatively weak in the northern part of the domain and largest in the eastern part of the domain (Fig. 4c,f,i), possibly due to the Kuroshio Current.

We next examined the shape and scale parameters of the Weibull fit to the surface current of Nan-Wan Bay (Fig. 4). As expected, the spatial pattern of the scale parameter, $\lambda$, is similar to the pattern of the current speed (the second column of Fig. 4 versus the third column). Interestingly, the shape parameter of the Weibull distribution, $k$ (first column Fig. 4), exhibits large spatial and seasonal variability. While the $k$ parameter is maximal in the northwestern region during the winter, it is maximal in the southeastern region during the summer, suggesting a different dynamics during these two seasons, probably related to tides as these also exhibit strong seasonal variations. Moreover, the $k$ parameter is much larger during the summer ($k \approx 2.21$) than during the winter ($k \approx 1.66$ Table 1) and exhibits large spatial variability in both seasons (from 1.8 to 2.8 during the summer and from 1.5 to 2 during the winter). This is probably related to the tides that are stronger during the fall and summer periods; more accurately, the spatial mean standard deviations of the tidal component were 21.5, 25.5, 30.85, 23.6, and 18.9 cm s$^{-1}$ for the annual, summer, fall, winter, and spring periods, respectively [note that the fall value is less reliable due to lack of data.] Note that the annual spatial mean standard deviations is necessarily equals to the mean of the spatial means of all seasons.



A comparison of the results shown in Fig. 4 with the ones from the Gulf of Elat (Fig. 3 of Ashkenazy and Gildor, 2011) indicates a similar range of spatial and seasonal variability. Yet the surface currents in the Gulf of Elat are much weaker than those in Nan-Wan Bay and the value of the shape parameter, $k$, is lower there. We thus conclude that the spatial and seasonal variability of the surface currents' statistical parameters is not unique to Elat and, indeed, is also present in the quite different region and environment of Nan-Wan Bay. We stress here that we observed the spatial variability, both in Nan-Wan Bay and the Gulf of Elat, in a relatively small region on the order of ten by ten kilometers.

Encouraged by the apparent asymmetry of the surface speed (see the black arrows in Fig. 3a), we next studied the temporal asymmetry of the surface current field, both in Nan-Wan Bay and the Gulf of Elat, following Eq. (2). Fig. 5a depicts the Nan-Wan Bay asymmetry measure, $A(\tau)$, of the annual time series for a one hour time interval. First, we found large spatial variability in $A(\tau)$, in which it is less than 0.5 in parts of the northern side of the basin and larger than 0.5 in the southern side of the domain. This indicates that statistically the current speed increases gradually and decreases more rapidly in the southern part of the domain, in contrast to the current surface increase/decrease in extensive regions in the northern side of the basin. The asymmetry range is between $0.48 < A < 0.52$, indicating up to 4% positive/negative steps compared to negative/positive steps. Second, we randomly shuffled the time series of each grid point, calculated the asymmetry measure $A$ and found the mean value $\pm$ 1 std of $A = 0.5 \pm 0.003$. This indicates that the reported asymmetry is significant, and it is well outside these error bars. Third, for a time lag of $\tau = 7$ hours (Fig. 5b), we found that the asymmetry changes sign for extensive parts of the domain. This suggests that the asymmetry is linked to tides that have a semi-diurnal periodicity for which the asymmetry should change sign for a time lag that is half the period. Fourth, the asymmetry during the summer (Fig. 5d) was found to be much stronger than the winter one (Fig. 5c), again suggesting that the asymmetry is influenced by the tides, as the tidal signal is stronger during the summer than during the winter. In the summer, $A$ exceeds 0.54, indicating that in the southern part of the domain, there are at least 8% more positive increments of the surface currents than negative ones. Finally, the pattern of the annual asymmetry field (Fig. 5a) was found to be similar to the summer one (Fig. 5d).

We performed a similar asymmetry analysis for the Gulf of Elat's surface currents (Fig. 6). Here, however, we did not find significant asymmetry as $A$ fluctuated very closely around 0.5 with an asymmetry similar to a randomly shuffled time series. This absence of asymmetry in the Gulf of Elat suggests that it has a weak tidal signal, as well as supporting the proposition that the surface current asymmetry is connected to the tides.

To verify the hypothesis that the tides underlie the observed asymmetry in Nan-Wan Bay, we decomposed the surface current time series, at every grid point in the domain, into tidal and detided components. In Fig. 3a,b, we present this decomposition, which indicates a large and apparently asymmetric tidal component. We then calculated the asymmetry of the tidal and detided components of the surface currents (Fig. 7). While the detided signals exhibit hardly any asymmetry (Fig. 7b,d,f) the tidal component is highly asymmetric (Fig. 7a,c,e), even more than the original time series. Also here the asymmetry during the summer is much stronger than the winter one. The range of asymmetry during the summer is $0.44 < A < 0.56$, indicating that the relative number of positive/negative increments in the tidal component of the surface currents can reach 12%. We thus conclude that the observed asymmetry in Nan-Wan Bay (both spatial and temporal as shown in Fig. 5) is due to the asymmetry of the tides in the region.



## 5 Summary

We analyzed the statistical properties of CODAR-based surface current fields in Nan-Wan Bay, Taiwan, and compared them to the statistical properties of CODAR-based surface current fields from the Gulf of Elat, Israel. The study area of both areas is on the order of $10 \times 10$ km. We fitted the PDFs of the surface currents to the Weibull distribution and found large spatial and seasonal variability of the Weibull distribution parameters (the shape $k$ and scale $\lambda$ parameters) in both basins, in spite of the many differences between the two regions. In addition, we also analyzed the temporal asymmetry of the surface current time series and found that Nan-Wan Bay's currents are asymmetric, while those of the Gulf of Elat are not. The asymmetry in Nan-Wan Bay is stronger in the summer. By analyzing the asymmetry of the tidal component of the currents, we associated the observed asymmetry with the tides.

Many previous studies have reported on the asymmetry of tidal currents, mainly in estuaries and coastal regions (e.g., Boon, 1975; Friedrichs and Aubrey, 1988; Blanton et al., 2002; Wang et al., 2002; Hoitink et al., 2003; Nidzieko, 2010). The tidal asymmetry is influenced by, among other factors, the bathymetry of the basin. The spatial differences in the asymmetry may be related to differences in the bathymetry of the basin, to the regional variability of the flow pattern, and to the asymmetry between flood-ebb tidal currents (Lee et al., 1999a); we hope to explore these possibilities in the near future. In addition, the asymmetry of satellite-based surface current data may be studied in the future as the temporal and spatial resolutions of this altimetry-based data constantly increase with time.

*Acknowledgements.* The authors thank the Israel-Taiwan Ministry of Science and Technology Program for financial support.



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



**Table 1.** Summary of the surface current statistics of Nan-Wan Bay. The table includes: the shape parameter $k$ of the Weibull distribution, the scale parameter $\lambda$ (cm s$^{-1}$) of the Weibull distribution, the mean zonal current $u$ (cm s$^{-1}$), the mean meridional current $v$ (cm s$^{-1}$), and the mean current speed $\sqrt{u^2+v^2}$ (cm s$^{-1}$). The mean $\pm$ one standard deviation is given for the annual, winter and summer periods.

| Parameter | Annual | Winter | Summer |
|---|---|---|---|
| $k$ | $1.78 \pm 0.09$ | $1.66 \pm 0.11$ | $2.21 \pm 0.03$ |
| $\lambda$ | $74.1 \pm 12.2$ | $67.4 \pm 10.8$ | $90.6 \pm 18.7$ |
| $u$ | $32.6 \pm 15.8$ | $16 \pm 14.4$ | $67.1 \pm 25.1$ |
| $v$ | $-2.16 \pm 9.32$ | $-9 \pm 7.56$ | $2.27 \pm 16.8$ |
| $\sqrt{u^2+v^2}$ | $65.9 \pm 10.8$ | $60.3 \pm 9.77$ | $80.43 \pm 16.7$ |





**Figure 1.** (a) Southeastern China-Taiwan region. The white rectangle indicates the region of Nan-Wan Bay. (b) Northern Red Sea region. The white rectangle indicates the northern part of the Gulf of Elat. (c) The bathymetry of Nan-Wan Bay, marked by the white rectangle in panel (a). The white rectangle indicates the approximate regions covered by the CODAR stations. (d) The bathymetry of the northern Gulf of Elat, marked by the white rectangle in panel (b). The white rectangle indicates the approximate regions covered by the CODAR stations.



**Figure 2.** Current field of Nan-Wan Bay as estimated by the CODAR. (a) Snapshot from 19:00 on March 7, 2013. (b) Annual mean current field. (c) Winter (DJF) mean current field. (d) Summer (JJA) mean current field. The filled red circles indicate the locations of the CODAR stations. The "x" indicates the location of the sample time series shown in Fig. 3a.



**Figure 3.** (a) A sample of speed time series (blue) spanning July 12, 2013 to August 1, 2013, from $120°$E, $21.87°$N (indicated by the "x" in Fig. 2) and its corresponding detided time series (red). (b) The tidal component of the current speed time series shown in (a). (c) The probability density function (PDF) of the current speed time series (filled blue circles) of JJA 2013 (part of which is shown in (a)), its best Weibull distribution fit (red curve) and the Weibull PDF for $k = 1$. (d) Same as (c) in a semi-log presentation.



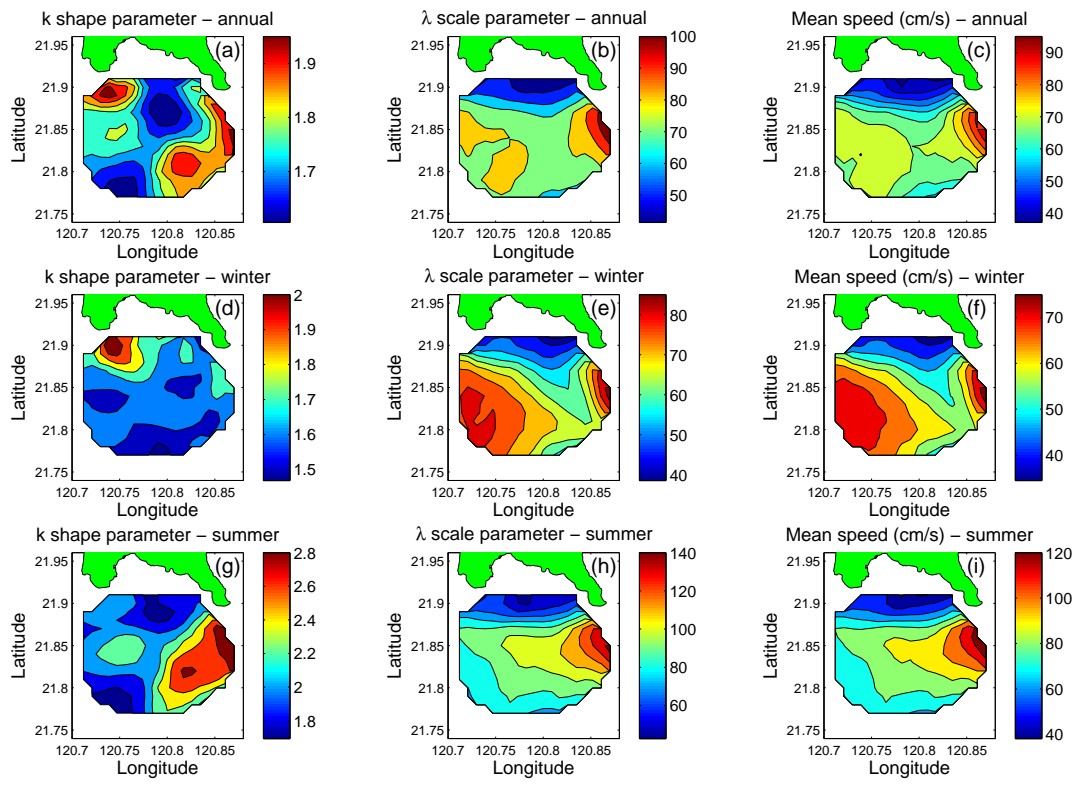

**Figure 4.** Summary of the statistical results. Left column: the shape $k$ parameter of the Weibull distribution; middle column: the scale $\lambda$ parameter of the Weibull distribution; right column: mean current speed (in cm/s). The first, second and third rows summarize the annual (1-3-2013 to 1-3-2014), winter (DJF of 2013-2014) and summer (JJA, 2013) results.





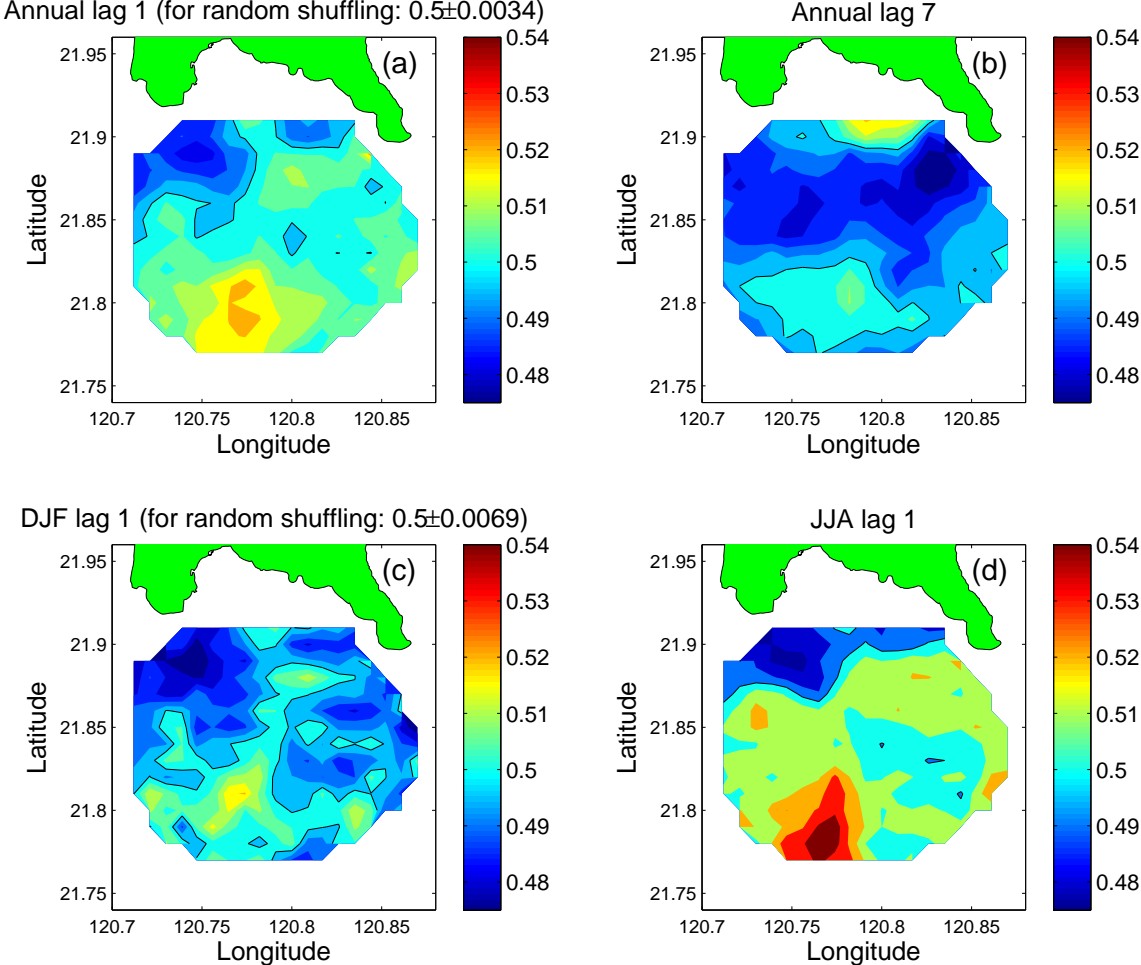

**Figure 5.** The asymmetry ratio [Eq. (2)] for the (a) 1 hour time interval (lag) for annual time series, (b) 7 hours time interval for annual time series, (c) 1 hour time interval for winter time series, and (d) 1 hour time interval for summer time series. The 0.5 value is indicated by the black contour line.





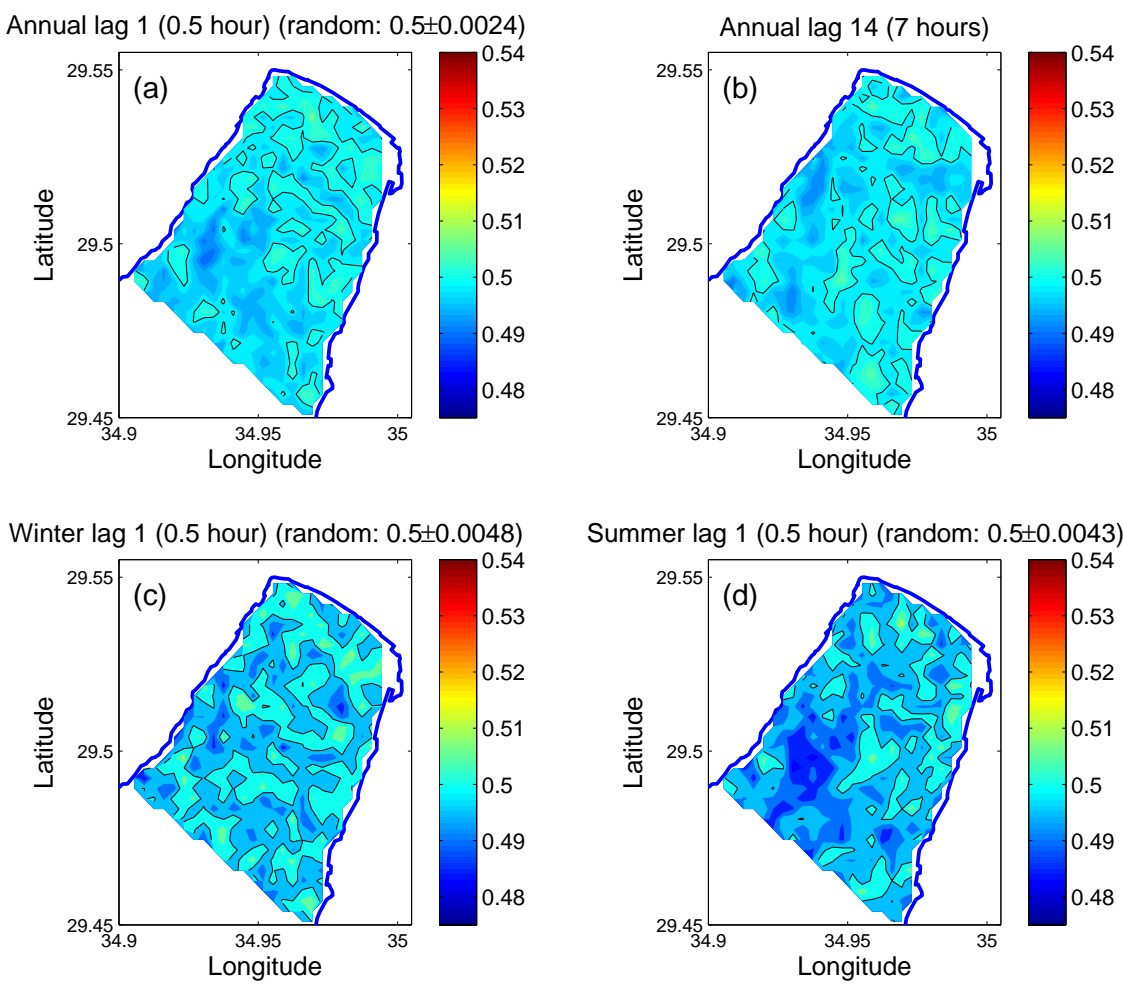

**Figure 6.** Same as Fig. 5 for the Gulf of Elat. No significant asymmetry is observed.





**Figure 7.** The asymmetry ratio for the annual (upper panels), winter (middle panels), and summer (bottom panels) time series, and for the tidal (left panels) and detided (right panels) time series. The 0.5 value is indicated by the black contour line.