# Peer review of "Current temporal asymmetry and the role of tides: Nan-Wan Bay vs. the Gulf of Elat"

_Ocean Science, 2016_

## Referee Comment (RC1) · Anonymous Referee #1 · 6 Apr 2016

**General comments**

The authors present an observational study into the statistical current properties of two profoundly different bays around the world. In brief, they find similarity in the ranges of spatial and seasonal variability, but large differences in the temporal asymmetries. In my opinion, the paper is in principle clearly written, regarding methods and results. However, to truly appreciate the added value of this study, I think the motivation of the study should be better articulated (including the choice of bays) as well as the wider implications and overall significance of the results.

[Figure]

**Specific comments**

- **§1, first goal.** The first goal, stated on line 17 op page 2, only seems meaningful because of the outcome. I mean: what if you would have found the opposite results, i.e. different statistics? In my opinion, you could not conclude anything about uniqueness because of the profound differences between the two bays. Please comment on this.

- **§1, choice of bays.** These differences between the two bays appear to be so profound that it is hard to really learn from the results. To avoid the impression of a somewhat artificial choice, I invite the authors to better motivate their choice to compare these two bays.

- **§3.4, asymmetry.** The notation of $\tau$ seems incorrect. According to Eq.(2), it is a number of time steps, i.e. an integer number. Yet, according to the text it is a time interval, measured in days. What is missing is the conversion by the time step $\Delta t$ of the time series. Correct would be: time interval $\tau = N_\tau \Delta t$ with $N_\tau$ the number of time steps to be used in the summation in Eq.(2). Please correct/clarify.

- **§4, standard deviation.** I think the last statement on p.6 (line 33) is only correct if, in the calculation of the standard deviations per season, the annual mean is used (rather than the mean of that particular season). For example: it is theoretically possible to have zero standard deviations per season (constant values within season, but differing from one season to another), in combination with a nonzero overall (annual) standard deviation. Can you comment on this?

- **§4, summary.** I miss some elaboration on the wider implications of these results. This makes it hard for me to assess the overall significance of the results. Please expand.

**Technical corrections**

- Throughout manuscript: please be consistent with 'fall' vs 'autumn'.

- §2.1-2, study regions. Perhaps consider mentioning the form factor $F$ to quantify the relative importance of diurnal and semidiurnal tides for both basins?

- Page 5, below Eq.(1). Please state that $x$ represents the random variable (symbol not explained).

- Page 5, line 15. It is not clear that these are three alternatives: I guess *either* (i), (ii) *or* (iii) is used. Further to this, I presume that 'different moments' refers to statistical moments, and I think that 'hazard function' may not be clear to some readers.

- Page 5, line 32: "The asymmetry measure of the current speed..." change into "The asymmetry measure $A$ of the current speed..."

- Figures 3, 4, 5, and 7: please include in the caption that these plots are about Nan Wan Bay.

- Figure 5: it is not clear from the figure and caption that the quantity $A$ is plotted here. Please add.

Anonymous, 6 April 2016

---

## Referee Comment (RC2) · Anonymous Referee #2 · 8 Apr 2016

This paper makes some interesting and important contributions, in particular the intordution of the somewhat unusal distribution the authors promote; at least to me the distribution was not exaclty a household name. The paper is unquestionably competently done, and well presented.

There are two main issues I thought the authors can better address in a future review.

The first is the rationale for the comparison of the two bays. As I read the ms, my strong impression was that the comparison was made for one reason only: the presence of the radar system the authors use in both locales. This is not a complleing reason for comparison. On the contrary are more compelling arguments against the comparison, in particular that the bays are SO dramatically bathymetrically and geometrically different, and that they are forced by fundamentally different processes. These two

observations taken together suggest that neither bay is a particularly interesting or apt conparison for the other. To be sure, each is perfectly interesting in its own right. It's just the comparison that feels strained.

Second, I can think of many physical variables to characterize the flow by. Time of acceleration of surface current seems secondary at best, and the authors do not provide compelling rationale for this peculiar choice.

But - once they made the choice to compare those two fundamentally different bays, and once they chose the physical variable they did, the authors did an excellent job deonstrating a very useful and practical application for thse quite unique data sets.

———————————————

---

## Referee Comment (RC3) · Anonymous Referee #1 · 20 Apr 2016

It is a pleasure to see how the authors have responded to my first review. I am satisfied with the answers and improvements, although I did not manage to find an updated .pdf-version of the manuscript. So, on the basis of the authors' responses (and expecting that they will be reflected in the new manuscript), I am happy to recommend publication.

---

## Author Comment (AC1) · 20 Apr 2016

We thank Reviewer 1 for her/his thoughtful comments. Reviewer 1 concludes that *"In my opinion, the paper is in principle clearly written, regarding methods and results. However, to truly appreciate the added value of this study, I think the motivation of the study should be better articulated (including the choice of bays) as well as the wider implications and overall significance of the results."* We are grateful for the reviewer's comments which helped us to improve the paper. Below is our detailed response.

**Specific comments**

**§1, first goal.** The first goal, stated on line 17 op page 2, only seems meaningful because of the outcome. I mean: what if you would have found the opposite results, i.e. different statistics? In my opinion, you could not conclude anything about uniqueness

because of the profound differences between the two bays. Please comment on this.

We agree and rephrased the goal of the paper as follows:

> The first goal of the present study was to verify whether the spatial variability of the statistical properties of the surface currents in the Gulf of Elat, reported in Ashkenazy and Gildor (2011), also exists in a quite different environment such as the Nan-Wan Bay of Taiwan (Fig. 1a,c).

**§1, choice of bays.** These differences between the two bays appear to be so profound that it is hard to really learn from the results. To avoid the impression of a somewhat artificial choice, I invite the authors to better motivate their choice to compare these two bays.

These two bays are very different, as described in manuscript, while at the same time, high-quality radar data exist for both. This is exactly why we choose these two locations. It wouldn't be that interesting to compare two similar bays. Moreover, had the results show that both bays share the same statistics (in all aspects), it would have been very surprising. That some of the characteristics are the same despite the large differences between the bays, suggest that our choice was actually not so bad, and that the reported conclusion regarding the natural variability of the statistical properties of surface currents is relevant in other marine environments.

We stress that we purposely choose such different environments in page 8, lines 12-14. We also state in the Summary section:

> We fitted the PDFs of the surface currents to the Weibull distribution and found large spatial and seasonal variability of the Weibull distribution parameters (the shape $k$ and scale $\lambda$ parameters) in both basins, in spite of the many differences between the two regions.

**§3.4, asymmetry.** The notation of $\tau$ seems incorrect. According to Eq. (2), it is a

number of time steps, i.e. an integer number. Yet, according to the text it is a time interval, measured in days. What is missing is the conversion by the time step $\Delta t$ of the time series. Correct would be: time interval $\tau = N_\tau \Delta t$ with $N_\tau$ the number of time steps to be used in the summation in Eq. (2). Please correct/clarify., choice of bays.

We agree and changed the text and Eq. (2) accordingly as follows:

> The asymmetry measure, $A$, of the current speed time series $s_i$ ($i = 1 \ldots N$ where $N$ is the length of the time series) can be expressed as:
>
> $$2A(\tau) = \frac{1}{N - N_\tau} \sum_{i=1}^{N - N_\tau} \Theta(s_{i+N_\tau} - s_i), \tag{1}$$
>
> where $\tau = N_\tau \Delta t$ is the asymmetry time interval, $\Delta t$ is the measurement temporal resolution, $N_\tau$ is the number of time steps compose the asymmetry time interval, and $\Theta(x)$ is a step function which is 1 for $x > 0$ and zero otherwise.

**§4, standard deviation.** I think the last statement on p.6 (line 33) is only correct if, in the calculation of the standard deviations per season, the annual mean is used (rather than the mean of that particular season). For example: it is theoretically possible to have zero standard deviations per season (constant values within season, but differing from one season to another), in combination with a nonzero overall (annual) standard deviation. Can you comment on this?

Following the reviewer's comment we decided to remove this confusing sentence from the revised manuscript.

**§5, summary.** I miss some elaboration on the wider implications of these results. This makes it hard for me to assess the overall significance of the results. Please expand.

Following the reviewer's comment we added the following sentences to the end of the summary section of the revised manuscript:

Our results indicate large spatial variability of the statistical properties of surface currents, even in small regions of a few kilometers and in very different environments. Thus, regional ocean modeling verification as well as the estimation of kinetic energy that can be extracted using ocean currents should be performed by using sufficiently fine spatial resolution. In addition, the statistical characteristics of the various regions should be used as a benchmark for model performance.

**Technical corrections**

Throughout manuscript: please be consistent with 'fall' vs 'autumn'.

Done–we now only use the term 'fall'. Thank you.

§2.1-2, study regions. Perhaps consider mentioning the form factor $F$ to quantify the relative importance of diurnal and semidiurnal tides for both basins?

We agree that the form factor $F$ is a useful measure to quantify the importance of diurnal and semidiurnal tides. Yet, since it is clear that tides are much more significant in Nan Wan Bay compare to the Gulf of Elat, and to simplify and ease the reading of the manuscript, we prefer not to introduce and present this form factor measure.

Page 5, below Eq. (1). Please state that $x$ represents the random variable (symbol not explained).

We now write that $x$ is a "Weibull random variable".

Page 5, line 15. It is not clear that these are three alternatives: I guess either (i), (ii) or (iii) is used. Further to this, I presume that 'different moments' refers to statistical moments, and I think that 'hazard function' may not be clear to some readers.

We actually analyzed using all three alternatives. The different methods yielded similar results and we mention it in the text. Following the reviewer's comment we rephrased these sentences as follows:

...it is possible to estimate $k$ using either (i) the different statistical moments, or (ii) the hazard function (see Ashkenazy and Gildor, 2011, for more details), or (iii) the maximum likelihood estimator of the Weibull distribution .... The different methods yielded similar results, and we thus present below only the results that are based on the different moment estimation (see Ashkenazy and Gildor, 2011).

Page 5, line 32: "The asymmetry measure of the current speed..." change into "The asymmetry measure $A$ of the current speed..."

Done.

Figures 3, 4, 5, and 7: please include in the caption that these plots are about Nan Wan Bay

Done.

Figure 5: it is not clear from the figure and caption that the quantity A is plotted here. Please add.

Done.

---

## Author Comment (AC2) · 20 Apr 2016

We thank very much Reviewer 2 for the important comments. Reviewer 2 stated that *"This paper makes some interesting and important contributions, in particular the introduction of the somewhat unusal distribution the authors promote; at least to me the distribution was not exactly a household name. The paper is unquestionably competently done, and well presented.".* At the end of review, reviewer 2 wrote: *"But - once they made the choice to compare those two fundamentally different bays, and once they chose the physical variable they did, the authors did an excellent job deonstrating a very useful and practical application for these quite unique data sets."* We thank the reviewer for this evaluation and address below all the other comments of the reviewer.

There are two main issues I thought the authors can better address in a future review.

[Figure]

The first is the rationale for the comparison of the two bays. As I read the ms, my strong impression was that the comparison was made for one reason only: the presence of the radar system the authors use in both locales. This is not a complleing reason for comparison. On the contrary are more compelling arguments against the comparison, in particular that the bays are SO dramatically bathymetrically and geometrically different, and that they are forced by fundamentally different processes. These two observations taken together suggest that neither bay is a particularly interesting or apt conparison for the other. To be sure, each is perfectly interesting in its own right. It's just the comparison that feels strained.

These two bays are very different, as described in manuscript, while at the same time, high-quality radar data exist for both. This is exactly why we choose these two locations. It wouldn't be that interesting to compare two similar bays. Moreover, had the results show that both bays share the same statistics (in all aspects), it would have been very surprising. That some of the characteristics are the same despite the large differences between the bays, suggest that our choice was actually not so bad, and that the reported conclusion regarding the natural variability of the statistical properties of surface currents is relevant in other marine environments.

We stress that we purposely choose such different environments in page 8, lines 12-14. We also state in the Summary section:

> We fitted the PDFs of the surface currents to the Weibull distribution and found large spatial and seasonal variability of the Weibull distribution parameters (the shape $k$ and scale $\lambda$ parameters) in both basins, in spite of the many differences between the two regions.

Second, I can think of many physical variables to characterize the flow by. Time of acceleration of surface current seems secondary at best, and the authors do not provide compelling rationale for this peculiar choice.

We agree that are many other measures that one could use. Yet, we find the parameters of the probability density function as well as the asymmetry of the time series to be fundamental and interesting enough to be considered. Moreover, as now suggested in the Summary, these statistical characteristics can be easily used as a benchmark for model performance. We hope to explore more features of the time series in the future.
* * *

---

## Author Response (AR2)

Department of Solar Energy and Environmental Physics
The Jacob Blaustein Institutes for Desert Research
Ben-Gurion University of the Negev, Israel
ashkena@bgu.ac.il
Yosef Ashkenazy

May 12, 2016

Dr. John M. Huthnance
Handling Topic Editor, Ocean Science
jmh@noc.ac.uk

**RE: os-2016-8 "Current temporal asymmetry and the role of tides: Nan-Wan Bay vs. the Gulf of Elat"** by Yosef Ashkenazy, Erick Fredj, Hezi Gildor, Gwo-Ching Gong, and Hung-Jen Lee

Dear Dr. Huthnance,

Thank you very much for your detailed response on the revised manuscript. It is unusual that editors make the efforts to read carefully the papers they are considering and we really appreciate your efforts and helped us to improve our manuscript. Please find below our response to all your comments. In your report you wrote:

> Thank-you for your revised version. However, I need to tell you that referee 1 wanted "major revisions" and to see the revised version again. I think this means that they were expecting rather more revision. I guess that may be in respect of motivation and the wider significance (both reviewers only rated the significance as "fair"). So I am asking you please to consider doing rather more in response to (both) reviewers comments, so that I can sensibly send it back for re-review, at least to the one that asked for it.

We note that Reviewer 1 already responded to our reply and to the revisions we have performed in response to his/her comments and was actually satisfied with them. This Reviewer 1 response was posted in the on-line discussions web-page on 20 April, 2016. Reviewer's 1 second report is:

> It is a pleasure to see how the authors have responded to my first review. I am satisfied with the answers and improvements, although I did not manage to find an updated .pdf version of the manuscript. So, on the

basis of the authors responses (and expecting that they will be reflected in the new manuscript), I am happy to recommend publication.

We hope that you will find our revised manuscript suitable for publication in Ocean Science.

The revised version of the re-submission includes: our response to your comments, the (slightly) revised response to the reviewers' comments, the revised version of the manuscript, and the revised version of the manuscript that includes the differences between the initial submitted version of the manuscript and this revised version of the manuscript.

Looking forward to hearing from you.

Sincerely,

Yosef Ashkenazy, Erick Fredj, Hezi
Gildor, Gwo-Ching Gong, and
Hung-Jen Lee

**Editor**

**Specific comments**

Page 2 line 10. ". . properties of surface currents"?

Done.

Page 2 line 27. ". . would result in additional conclusions." ?

Done.

Page 3 line 5. ". . ("wan" is Chinese for bay) . ."

Done.

Page 3 lines 17-18. "The semidiurnal and diurnal tidal components of the currents in Nan-Wan Bay are approximately equal in their magnitude (Lee et al., 1997) but are modulated by the spring-neap tidal cycle. The temperature . ."

Done.

Page 3 line 27. The meaning of "semidiurnal peak" is not clear to me. Is it the maximum current within the 12-hour tidal cycle, or in the spring-neap cycle, or saying that semidiurnal exceeds diurnal? If this is clear I dont think you need to refer to the "form factor".

Following the Editor's suggestion we rewrote this sentence as follows:

> The semidiurnal peak, forced by the flux of water through the Straits of Tiran, dominates the weak diurnal tidal component . . .

Pages 6-7. There is a lot of emphasis on the asymmetry. In your response to referee 2 you say "we find . . to be fundamental and interesting enough to be considered". I can accept this but it would help if you stated more about why they are interesting in the revised text. I agree they could test model results. For example, there could be a relation with the degree of non-linearity and rectification (which models might find more difficult).

In response to the Editor comment we added the following sentences to the revised manuscript: on page 2, lines 29-31

> The asymmetry and other statistical characteristics of surface currents may be used to test and validate the performance of oceanic models.

and on page 7, lines 16-17

> The asymmetry measure may be used to test the performance of ocean models as it also quantifies nonlinear aspects of the underlying process.

 This sentence seems to say much the same thing twice. Also (line 34) "the surface current asymmetry" does not apply in the Gulf of Elat which the sentence is discussing, and (line 33) you dont need "suggests that it has a weak tidal signal" since you already know that the Gulf of Elat tides are weak.

We agree and rewrote this problematic sentence as follows:

> This absence of asymmetry is associated with the weak tidal signal in the Gulf of Elat.

 Twice you say in the response that "we stress that we purposely choose such different environments . ." but I dont really see this at page 8 lines 12-14 (I am not sure which version is referred to). Perhaps this could be added with reasons (why "purposely") in the appropriate place.

Probably the word "purposely" is not appropriate and we excluded it now from the response to the reviewers. As for the reference to "page 8, lines 12-14", mistakenly this was not in place. We rewrote our response to the reviewers as follows

> We highlight the choice of different environments in several places in the revised manuscript: (i) page 1, line 1-2, (ii) page 2, lines 17-19, (iii) page 2, lines 24-25, (iv) page 7, lines 11-13. In addition we also state in the Summary section (in page 8, lines 12-14):
>
> > We fitted the PDFs of the surface currents to the Weibull distribution and found large spatial and seasonal variability of the Weibull distribution parameters (the shape $k$ and scale $\lambda$ parameters) in both basins, in spite of the many differences between the two regions.

 Add "In addition, the statistical characteristics of the various regions should be used as a benchmark for model performance"? as in your response to reviewer 1 §5.

Done.

**Reviewer 1**

We thank Reviewer 1 for her/his thoughtful comments. Reviewer 1 concludes that *"In my opinion, the paper is in principle clearly written, regarding methods and results. However, to truly appreciate the added value of this study, I think the motivation of the study should be better articulated (including the choice of bays) as well as the wider implications and overall significance of the results."* We are grateful for the reviewer's comments which helped us to improve the paper. Below is our detailed response.

**Specific comments**

**§1, first goal.** The first goal, stated on line 17 op page 2, only seems meaningful because of the outcome. I mean: what if you would have found the opposite results, i.e. different statistics? In my opinion, you could not conclude anything about uniqueness because of the profound differences between the two bays. Please comment on this.

We agree and rephrased the goal of the paper as follows:

> The first goal of the present study was to verify whether the spatial variability of the statistical properties of the surface currents in the Gulf of Elat, reported in Ashkenazy and Gildor (2011), also exists in a quite different environment such as the Nan-Wan Bay of Taiwan (Fig. 1a,c).

**§1, choice of bays.** These differences between the two bays appear to be so profound that it is hard to really learn from the results. To avoid the impression of a somewhat artificial choice, I invite the authors to better motivate their choice to compare these two bays.

These two bays are very different, as described in manuscript, while at the same time, high-quality radar data exist for both. This is exactly why we choose these two locations. It wouldn't be that interesting to compare two similar bays. Moreover, had the results show that both bays share the same statistics (in all aspects), it would have been very surprising. That some of the characteristics are the same despite the large differences between the bays, suggest that our choice was actually not so bad, and that the reported conclusion regarding the natural variability of the statistical properties of surface currents is relevant in other marine environments.

We highlight the choice of different environments in several places in the revised manuscript: (i) page 1, line 1-2, (ii) page 2, lines 17-19, (iii) page 2, lines 24-25, (iv) page 7, lines 11-13. In addition we also state in the Summary section (in page 8, lines 12-14):

> We fitted the PDFs of the surface currents to the Weibull distribution and found large spatial and seasonal variability of the Weibull distribution

parameters (the shape $k$ and scale $\lambda$ parameters) in both basins, in spite of the many differences between the two regions.

**§3.4, asymmetry.** The notation of $\tau$ seems incorrect. According to Eq. (2), it is a number of time steps, i.e. an integer number. Yet, according to the text it is a time interval, measured in days. What is missing is the conversion by the time step $\Delta t$ of the time series. Correct would be: time interval $\tau = N_\tau \Delta t$ with $N_\tau$ the number of time steps to be used in the summation in Eq. (2). Please correct/clarify., choice of bays.

We agree and changed the text and Eq. (2) accordingly as follows:

> The asymmetry measure, $A$, of the current speed time series $s_i$ ($i = 1 \dots N$ where $N$ is the length of the time series) can be expressed as:
>
> $$A(\tau) = \frac{1}{N - N_\tau} \sum_{i=1}^{N - N_\tau} \Theta(s_{i+N_\tau} - s_i), \tag{2}$$
>
> where $\tau = N_\tau \Delta t$ is the asymmetry time interval, $\Delta t$ is the measurement temporal resolution, $N_\tau$ is the number of time steps compose the asymmetry time interval, and $\Theta(x)$ is a step function which is 1 for $x > 0$ and zero otherwise.

**§4, standard deviation.** I think the last statement on p.6 (line 33) is only correct if, in the calculation of the standard deviations per season, the annual mean is used (rather than the mean of that particular season). For example: it is theoretically possible to have zero standard deviations per season (constant values within season, but differing from one season to another), in combination with a nonzero overall (annual) standard deviation. Can you comment on this?

Following the reviewer's comment we decided to remove this confusing sentence from the revised manuscript.

**§5, summary.** I miss some elaboration on the wider implications of these results. This makes it hard for me to assess the overall significance of the results. Please expand.

Following the reviewer's comment we added the following sentences to the end of the summary section of the revised manuscript:

> Our results indicate large spatial variability of the statistical properties of surface currents, even in small regions of a few kilometers and in very different environments. Thus, regional ocean modeling verification as well as the estimation of kinetic energy that can be extracted using ocean currents should be performed by using sufficiently fine spatial resolution. In addition, the statistical characteristics of the various regions should

be used as a benchmark for model performance.

**Technical corrections**

Throughout manuscript: please be consistent with fall vs autumn.

Done–we now only use the term 'fall'. Thank you.

§2.1-2, study regions. Perhaps consider mentioning the form factor $F$ to quantify the relative importance of diurnal and semidiurnal tides for both basins?

We agree that the form factor $F$ is a useful measure to quantify the importance of diurnal and semidiurnal tides. Yet, since it is clear that tides are much more significant in Nan Wan Bay compare to the Gulf of Elat, and to simplify and ease the reading of the manuscript, we prefer not to introduce and present this form factor measure.

Page 5, below Eq. (1). Please state that $x$ represents the random variable (symbol not explained).

We now write that $x$ is a "Weibull random variable".

Page 5, line 15. It is not clear that these are three alternatives: I guess either (i), (ii) or (iii) is used. Further to this, I presume that different moments refers to statistical moments, and I think that hazard function may not be clear to some readers.

We actually analyzed using all three alternatives. The different methods yielded similar results and we mention it in the text. Following the reviewer's comment we rephrased these sentences as follows:

> ... it is possible to estimate $k$ using either (i) the different statistical moments, or (ii) the hazard function (see Ashkenazy and Gildor, 2011, for more details), or (iii) the maximum likelihood estimator of the Weibull distribution .... The different methods yielded similar results, and we thus present below only the results that are based on the different moment estimation (see Ashkenazy and Gildor, 2011).

Page 5, line 32: "The asymmetry measure of the current speed..." change into "The asymmetry measure $A$ of the current speed..."

Done.

Figures 3, 4, 5, and 7: please include in the caption that these plots are about Nan Wan Bay

Done.

Figure 5: it is not clear from the figure and caption that the quantity A is plotted here. Please add.

Done.

**Reviewer 2**

We thank very much Reviewer 2 for the important comments. Reviewer 2 stated that *"This paper makes some interesting and important contributions, in particular the introduction of the somewhat unusal distribution the authors promote; at least to me the distribution was not exactly a household name. The paper is unquestionably competently done, and well presented."*. At the end of review, reviewer 2 wrote: *"But - once they made the choice to compare those two fundamentally different bays, and once they chose the physical variable they did, the authors did an excellent job deonstrating a very useful and practical application for these quite unique data sets."* We thank the reviewer for this evaluation and address below all the other comments of the reviewer.

There are two main issues I thought the authors can better address in a future review.

The first is the rationale for the comparison of the two bays. As I read the ms, my strong impression was that the comparison was made for one reason only: the presence of the radar system the authors use in both locales. This is not a complleing reason for comparison. On the contrary are more compelling arguments against the comparison, in particular that the bays are SO dramatically bathymetrically and geometrically different, and that they are forced by fundamentally different processes. These two observations taken together suggest that neither bay is a particularly interesting or apt conparison for the other. To be sure, each is perfectly interesting in its own right. Its just the comparison that feels strained.

These two bays are very different, as described in manuscript, while at the same time, high-quality radar data exist for both. This is exactly why we choose these two locations. It wouldn't be that interesting to compare two similar bays. Moreover, had the results show that both bays share the same statistics (in all aspects), it would have been very surprising. That some of the characteristics are the same despite the large differences between the bays, suggest that our choice was actually not so bad, and that the reported conclusion regarding the natural variability of the statistical properties of surface currents is relevant in other marine environments.

We highlight the choice of different environments in several places in the revised manuscript: (i) page 1, line 1-2, (ii) page 2, lines 17-19, (iii) page 2, lines 24-25, (iv) page 7, lines 11-13. In addition we also state in the Summary section (in page 8, lines 12-14):

> We fitted the PDFs of the surface currents to the Weibull distribution and found large spatial and seasonal variability of the Weibull distribution parameters (the shape $k$ and scale $\lambda$ parameters) in both basins, in spite of the many differences between the two regions.

Second, I can think of many physical variables to characterize the flow by. Time

of acceleration of surface current seems secondary at best, and the authors do not provide compelling rationale for this peculiar choice.

We agree that are many other measures that one could use. Yet, we find the parameters of the probability density function as well as the asymmetry of the time series to be fundamental and interesting enough to be considered. Moreover, as now suggested in the Summary, these statistical characteristics can be easily used as a benchmark for model performance. We hope to explore more features of the time series in the future.

[revised manuscript text omitted]

---

## Author Response (AR3)

Department of Solar Energy and Environmental Physics
The Jacob Blaustein Institutes for Desert Research
Ben-Gurion University of the Negev, Israel
ashkena@bgu.ac.il
Yosef Ashkenazy

May 18, 2016

Dr. John M. Huthnance
Handling Topic Editor, Ocean Science
jmh@noc.ac.uk

**RE: os-2016-8 "Current temporal asymmetry and the role of tides: Nan-Wan Bay vs. the Gulf of Elat"** by Yosef Ashkenazy, Erick Fredj, Hezi Gildor, Gwo-Ching Gong, and Hung-Jen Lee

Dear Dr. Huthnance,

Thank you very much for your last response on our manuscript. Following your comments we adopted the European style and replaced "meter", "kilometer", and "millimeter" by "metre", "kilometre", and "millimetre". We have also replaced (page 7, line 22) "and it is" with "being", as you suggested.

The revised version of the re-submission includes the revised version of the manuscript, and the revised version of the manuscript that includes the differences between the previous submitted version of the manuscript and this revised version of the manuscript. We hope that you will be able to accept the manuscript for publication in OS.

Looking forward to hearing from you.

Sincerely,

[revised manuscript text omitted]